# The impact of insurance institutional investors on corporate value from selection and creation perspective

Xing Rong[1], Tingting Zhang[1], Kai Liu[2,3,4]*

**1** School of Insurance, Southwestern University of Finance and Economics, Chengdu, China, **2** School of Mathematical and Computational Sciences, University of Prince Edward Island, Charlottetown, Canada, **3** Faculty of Computer Science, Dalhousie University, Halifax, Canada, **4** Big Data Research Lab, University of Waterloo, Waterloo, Canada

* kailiu@upei.ca

## Abstract

The majority of insurance investment funds are derived from policy liability debt funds. It differs from other institutional investors in a number of ways, including investment size, horizon, duration, risk, and so on. However, only a small portion of the extant literature focuses on in-depth and extensive analysis of Insurance Institutional Investors' holdings (IIIs). This study analyses the impact of shareholding by insurance institutions on the value of Shanghai and Shenzhen A-share listed companies in China's capital market. The paper offers three major contributions. First, we discovered that long-term equity-holding IIIs have both value selection and value creation functions. Second, the value creation function becomes more significant among long-term stock-holding IIIs with an increase in the period during which they retain the company's shares; Third, fast-in and fast-out (FIFO) IIIs have a value-inhibiting effect on the held company and serve a value selection role, rather than a value creation function. This study provides more insight on the lack of academic interest in insurance institutions and serves as a foundation and reference for the design of regulatory policies for insurance institutions' involvement in stock markets. It also gives empirical evidence for corporations to accurately analyze shareholding by insurance institutions. Furthermore, since this study concentrates on China's capital market, it can serve as a benchmark for other nations, particularly, those designated as developing market economies.

## Introduction

Insurance institutions maintain a position that cannot be overlooked as key institutional investors in capital markets across the globe. According to the data of China insurance Yearbook, the insurance industry accounted for 17.8% of the global financial industry's total assets in the fourth quarter of 2019. US insurance institutions had $6.5 trillion in financial assets and held more than 25% of US corporate bonds in 2017 [1]. Insurance funds, unlike other institutional investors, are mostly debt funds fueled by policy obligations, rather than self-owned investment funds [2]. Thus, the size, horizon, duration, and risk of insurance funds are determined

**Data Availability Statement:** The data that support the findings of this study are openly available in the Wind Financial Database at (http://www.wind.com.cn), RESET Database at (https://db.resset.com)

and CSMAR Database at (http://www.gtadata.com).

**Funding:** T.Z., Grant No. 71902163, the National Natural Science Foundation for Young Scientists of China, http://www.nsfc.gov.cn/. The funders had no role in study design, data collection and analysis, decision to publish, or preparation of the manuscript. T.Z., Grant No. 2019040027 and the Humanities and Social Sciences Youth Foundation of Ministry of Education of China,http://www.moe.gov.cn/s78/A13/.The funders had no role in study design, data collection and analysis, decision to publish, or preparation of the manuscript.

**Competing interests:** The authors have declared that no competing interests exist.

by the characteristics of the funds held, which must be matched with the insurance holders' liabilities [3]. Their impact on investee companies is likely to differ from that of other institutional investors due to these unique characteristics. The controversy over Insurance Institutional Investors' holdings (IIIs) controlling the Chinese real estate company, Vanke Group over the nearly 3-year long period from 2015 to 2017, led to insurance funds being dubbed "barbaric funds" in China, and sparked interest and discussion about IIIs. This negative impression regarding IIIs have not yet been overcome. Regrettably, most extant academic circles conduct research on institutional investors as a whole, with few focusing on the specific impact of IIIs' shareholding on company value.

Institutional investors are a significant source of capital with significant financial strength and investment potential [4], and a large body of research examines whether institutional investor stockholding can increase the value of the company owned. The available literature on institutional investors' role can be divided into three groups. (1) Institutional investors, according to one viewpoint, will actively participate in corporate governance, and thereby effectively boosting company performance and value [5–8]. According to this view, institutional investors can effectively supervise the company's investment operations [9], improve the quality of earnings [10–13], or cut agency costs by overseeing the behavior of major shareholders and executives of publicly traded companies [7, 14–16], which improve company performance and increase corporate value, as they have the advantages of abundant capital, high shareholding ratio, abundant information channels, and strong investment and research teams [17]. (2) However, others argue that institutional investor holdings have a harmful impact on corporate governance. They argue that institutional investors and company executives may work together to diverge from the goal of increasing business value [18–20] as they may be focused on short-term performance measures driven by the financial market, which can easily lead to a variety of issues in the company's long-term management [20]. (3) Finally, there is a belief that institutional investors are poorly controlled and that their participation has little impact on the value of a company. This view is suggested by Wahal [21] in their analysis of the influence of pension funds' active participation in corporate governance. Similarly, Pouraghajan et al. [22] stated that institutional investors in developing countries cannot fulfil the function of corporate governance monitoring. Therefore, the literature on the impact of institutional investors' holdings on firm value lacks consensus. Furthermore, despite several studies investigating the heterogeneity of institutional investors and discovering that securities investment funds, QFIIs, and social security funds differ in their enthusiasm for participating in corporate governance, value selection, or value creation roles, there is limited research on the heterogeneity of institutional investors [4, 23–27].

Regulators promote and encourage insurance institutions to undertake long-term investments and lead investment ideas due to the long-term stability of insurance funds. However, there are several concerns in the extant literature that have not been thoroughly investigated. For example, are insurance companies, committed towards making long-term investments? How will the presence of IIIs in a company's stock affect its value, and should firms be concerned about the presence of insurance institutions in their stock? Similarly, despite insurance institutions' funds being stable, their use is constrained by compensation and maturity payment obligations. Will insurance companies choose FIFO investments for short-term gain in this case? Is there a difference in value creation and discovery among insurance institutional investors with various investment horizons? This article delves into the impact of insurance institutions' stock ownership on a company's value to fill in some of the research gaps on these topics.

This article examines data from insurance institutions investing in Shanghai and Shenzhen A-share listed businesses in the Chinese capital market over the course of 25 quarters, from the

first quarter of 2012 to the first quarter of 2018. The samples of companies are classified into two categories based on the investment horizon: long-term shareholding and FIFO. The value selection and value creation models are used to investigate the impact of IIIs' stock ownership on corporate value. Theoretical analysis and empirical testing are followed by a robustness test and a breakdown of short-term and long-term holdings in this article. The study location was chosen as China for two reasons: First, China has the world's second-largest insurance market, behind only the United States. Simultaneously, China's insurance market has been ranked first in terms of growth rate among the top 20 insurance markets in the world from 2011 to 2020. Additionally, Chinese insurance funds' investments in A-shares accounted for 3.44 percent of the total market value at the end of the third quarter of 2019, and their bond investments accounted for 6.49 percent of the total size of China's bond market, making them the second largest institutional investor in the Chinese market after public funds. Second, as a growing economy, compared with the developed countries, China's insurance industry started later but experienced a rapid development in more than twenty years. Therefore, it is necessary to strengthen the insurance investment system as well as the regulatory environment. IIIs investment in China's largest real estate company, Vanke Group, drew the attention and debate of insurance institutional investors throughout China. Holdings in insurance funds were traditionally deemed as "barbaric funds," and were not desired by corporations. The communities' dreadful imprint on IIIs has yet to be erased. Therefore, this article uses China as an example to investigate the impact of IIIs' stock ownership on firm value. This can provide proposals for resolving difficulties and issues and serve as a reference for other countries around the world, particularly emerging economies with insufficient insurance systems [28].

The study discovered the following: Firstly, long-term stock-holding IIIs have both value selection and value creation capabilities. Second, among long-term shareholding IIIs, the value creation function increases in significance with an increase in the period of holding shares. Third, FIFO IIIs have a negative impact on the held company's value. They do not have the ability to create value, but they do have the ability to select value. The following aspects highlight the contributions of this paper: This paper undertakes a rigorous and in-depth analysis of the impact of IIIs' shareholding on company value, and rectifies the absence of research on IIIs to some extent. Insurance companies were used as an example by Ge and Weisbach [1] to ascertain whether the financial health of institutional investors has an impact on their investment portfolios. However, despite insurers being major and influential investors, there is little research on the impact of institutional investors on firm value. This study investigates the value selection and creation functions of institutional investors' holdings on enterprises from both a theoretical and empirical perspectives, and thereby expands previous research on institutional investor heterogeneity.

Second, this paper examines the motivation and ability of insurance companies to create value from both positive and negative motivations, and the characteristics of insurance funds, insurance businesses, new accounting standards, and the impact of various stages of insurance company development on value creation. It examines numerous considerations and choices of insurance fund shareholding on a theoretical level and demonstrates the need for incorporating institutional heterogeneity in more in-depth research of institutional investors' impact on company value. Theoretical analysis gives a reference point for businesses to assess III motivation.

Third, the value creation role is distinguished from an investment horizon in this study, and it thereby provides an empirical basis for regulatory authorities to encourage long-term holdings by insurance firms on a practical level. This article disproves the notion that insurance funds are "barbaric funds" and provides factual support for Chinese regulators' long-term investment strategy for insurance businesses. The empirical investigations serve as a

foundation and point of reference for the creation of regulations. The focus of this article on the impact of Chinese insurance institutions' shareholdings on firm value can be useful as a benchmark for other nations, particularly emerging economies.

## Theoretical analysis and hypotheses

### The value selection function of III

Insurance firms and insurance asset management businesses that invest in stocks are referred to as IIIs. The primary source of funds for IIIs is debt funds derived from policy liabilities. When holding shares, IIIs must assess if investment returns can sustain the operation of insurance businesses [1]. That is, the return on investment should be in line with the needs for profitability, liquidity, and safety. Consequently, whether it is short-term stock holding for profit or long-term stock holding to meet liabilities, insurance institutions must choose firms with investment value. This paper defines value selection as the insurance companies are able to select companies with development potential that will grow in value in the future to benefit shareholders. Many studies have revealed that institutional investors, including IIIs, have the ability and inclination to choose value. Studies such as Aggarwal et al. [29], Li and Li [19], Mu and Zhang [30], Shi and Li [31], and Wu and Jiang [32] have demonstrated that ratio of institutional investors' shareholding, and indicators like firm performance and market value have a substantial positive link. According to Wang and Ge [33], the shareholding ratio of life insurance businesses is substantially connected with the operating success of publicly traded companies. Wen and Feng [34] pointed out that securities' investment funds are under pressure to frequently publish investment results while discussing the relationship between heterogeneous institutions and independent innovation. However, as insurance funds are exempt from disclosing their investment portfolios on a regular basis and from participating in general fund rankings, they are incentivized to seek out more creative businesses to invest in. The ability of insurance institutions to select value is very clearly demonstrated in practice. The real estate industry, which was traditionally seen as a safe investment by insurance capital in China, was quickly abandoned by the end of 2021 as the real estate environment deteriorated and affected revenue. IIIs considered real estate businesses as "sweet pastry" in the past because of their high return on equity (ROE) and low volatility, and features such as value and growth. The insurance industry also intersects with the real estate market in terms of pension real estate. However, IIIs withdrew their money rapidly when the period of huge payouts in the real estate business ended. Therefore, choosing organizations that are worthy investments is an unavoidable requirement for insurance companies. In addition, insurance funds in China's capital market are huge in scale, have a lot of information channels, and have good screening skills [17]. This aids in value selection. Insurance institutions' shareholding scale continues to rise in the Chinese market, with the percentage of investments in the A-share circulating market rose from 3.6% in 2016 to 5.45 percent in 2018. The amount of money that insurance funds invested in stocks and securities accounted for 12.39 percent of the total at the end of the first quarter of 2019. This growth has been fueled in part by the China Banking and Insurance Regulatory Commission's suggestion to insurance funds to grow their equity holdings in publicly traded companies. Thus, insurance institutions have a great screening power to select desirable enterprises for investment based on government assistance and their own strength.

Based on the above analysis, this paper proposes the following first hypothesis:

**Research hypothesis H1**: IIIs with long-term or short-term shareholding have the ability to choose value, that is, the shareholding ratio of IIIs has a significant positive impact on company value.

## The value creation function of FIFO IIIs

Value creation refers to the increase in the value of the company held by the insurance company after the insurance company holds shares or increases their original holdings. The question of whether IIIs have a value creation function needs to be investigated further, and any discussion of this topic must take into account the holding term of IIIs. Institutional investors may have various long- and short-term goals, concerns, and effects on firm value. In this article, institutional investors who hold short-term equities are referred to as FIFO institutional investors, which signifies that these funds withdraw their money from stocks after a short period of time. Many studies suggest that institutional investors with short-term stock holdings are less motivated to conduct corporate monitoring tasks, and that they anticipate corporate agents to prioritize present success over long-term value, resulting in them being value selectors rather than value producers [23, 29]. Thus, for example, management may cut R&D expenditures in a firm with a high proportion of short-term institutional investors' holdings to avoid a decrease in the stock price [35]. Therefore, FIFO investors take on the role of short-term speculators, and often take the negative option of "voting with their feet" if they are displeased with the company's performance [36].

Insurance funds are characterized as the "stabilizer" of the capital market because of their unique benefits of huge scale, steady sources, and the pursuit of secure income. Thus, countries urge insurance funds to make long-term investments to support the capital market's stability. However, short-term holdings, are appealing to insurance firms as policyholder obligations are the primary source of insurance funding. Property insurance companies, for example, are generally short-term businesses, with insurance periods ranging from one day to one year. Short-term stock holdings make it easier to meet insurance firms' liquidity needs for insurance claims or maturity payments According to the extant literature, short-term IIIs' stock investments are short-sighted, and thus cannot improve company performance [37–40]. Furthermore, insurance businesses have a limited risk tolerance, and even short-term investments tend to diversify to achieve consistent but poor returns. Additionally, they have a low motivation and ability to participate in corporate governance [27, 41].

Based on the above analysis, this paper proposes the second hypothesis:

**Research hypothesis H2**: FIFO IIIs do not have the function of value creation, that is, FIFO IIIs' shareholding changes have no positive relationship with company value changes.

## The value-creating function of long-term equity-holding IIIs

The value-creating function of long-term holding institutional investors has been researched extensively in the literature (including IIIs). Institutional investors that hold a firm's stock for a longer period of time can provide ongoing and effective oversight and help the company operate better [42]. Long-term and heavy-holding institutional investors have a beneficial governance effect on the holding company, as demonstrated by Borokhovich et al. [43] and Brickley et al. [44] from the anti-takeover proposal, Bushee [35] from the investment decision of R&D expenses, and Almazan et al. [45] and Hartzell and Starks [46] from the standpoint of CEO remuneration, respectively. Chen et al. [23] used corporate Mergers and Acquisitions (M&A) decisions to reflect the degree of shareholder supervision over executives, and empirical research found that when a company faces difficulties, its long-term institutional investors usually take positive actions to help it weather the storm, resulting in significantly improved corporate performance. Bamahros and Wan-Hussin [47] showed that that there was an increase in the tendency of investors to provide oversight and promote the company's long-term worth by acting as a supervisor, with an increase in the holding time.

The impact of insurance institutions' long-term shareholding on company value is potentially two-sided. Insurance companies have a vested interest in creating value. First, the long-term holding demand of insurance companies and the poor risk tolerance of debt management push them to participate in corporate governance, supervise executives, reduce business agency expenses, and promote corporate value improvement to reap the long-term benefits. Liu et al. [2] evaluated the shareholding characteristics and preferences of insurance organizations and observed that IIIs have a long-term value investing philosophy, which is beneficial to a stable capital market. Second, the healthy development of insurance companies requires support from non-insurance businesses both upstream and downstream. Banks play a key role in the marketing of insurance goods as a part-time insurance agency channel. Insurance companies listing banks will have an impact on bank decision-making, allowing insurance businesses to benefit from synergies. The pharmaceutical and biological industries are inextricably linked to the cost and quality of medical insurance and critical illness insurance provided by insurance companies. The real estate sector is intertwined with the insurance industry's pension community, as well as pension and long-term care insurance. Furthermore, insurance firms are expected to offer not only current and future pensions, but also limited pension services that can be purchased by pensioners.

The auto sector is inextricably linked to insurance firms' auto insurance businesses. Insurance companies are expanding their industrial chain in sectors such as pre-sale, after-sale maintenance, daily maintenance, roadside assistance, and vehicle finance. Insurance companies are willing to offer more diverse products and services to their customers, which can increase customer contribution value; insurance companies make full use of the synergies among their various businesses in terms of brands, channels, funds, and so on, which can not only expand the customer base but also increase customer stickiness through cross-selling. To summarize, insurance companies are the final payers in a variety of industries, with a large client base and on-the-ground teams. They have inherent advantages in terms of location as well as significant economic benefits in terms of increasing business boundaries along the industry chain. As a result, we believe that insurance businesses have the desire to create value to benefit from strategic synergy. Third, the new accounting standard IFRS9 (International Financial Reporting Standards), has been fully implemented from 2021, and insurance companies are eager to boost long-term equity investment to reduce profit volatility. Equity assets must be valued at fair value under the new standard, and equity changes will be directly reflected in the income statement, and thus have a rather considerable influence on net profit. According to IFRS9, investments holding more than 5% of a company and a director seat can be included in the long-term equity investment account, which is accounted for using the equity method, so that investment performance is not affected by stock price fluctuations and is more stable. This has an impact on the evolution of insurance fund investment behavior toward long-term value generation to some extent.

Long-term IIIs, on the other hand, have negative incentives to build value, as creating value for companies necessitates an investment of time and money. Since insurance funds are commercial capital, they have no obligation to produce value for other businesses. The benefits of value creation does not always outweigh the expenses. Second, insurance companies are "stress-sensitive" institutional investors [23], as insurance companies and investee companies have commercial links [48], such as offering insurance services [49]. They are concerned that their own oversight and involvement in the enterprise's management may result in the loss of existing or potential commercial contracts and impact their own income. Therefore, the cost of performing its supervisory function is higher for insurance companies than for other pressure-resistant institutional investors (funds, social security funds, QFII, enterprise annuities), and it lacks the motivation to actively participate in corporate governance [50], making it

difficult to contribute to the creation of value for businesses. Third, the insurance industry's development and asset management are still in their infancy; there are variances in the development of insurance institutions, and not every insurance business is at the stage of expanding the insurance industry chain. Short-sighted behavior and the objective development stage are highly likely to result in no value generation function.

Based on the above analysis, this paper proposes a third hypothesis:

**Research hypothesis H3a**: Investors in long-term holding type insurance institutions have the function of value creation, which means that increases in their shareholding have a positive impact on the company's value changes.

**Research hypothesis H3b**: Long-term holding type insurance institutional investors do not have the function of value creation, i.e., shareholding changes of long-term holding type insurance institutional investors have no substantial association with changes in the company's value.

## Model design and variables

**Main research model design.**   Given that IIIs' decision to buy stocks in period t is based on the company's value performance in period t and before, and that IIIs' ability to choose value is reflected in the company's performance after period t (where t represents time; since the data used in this article spans quarters, t represents a quarter, where t = 1, 2, 3. . . . 25). Furthermore, there may be a strong relationship between insurance institutions' shareholding ratio and company value: insurance institutions' participation in corporate governance may promote better value performance of the company; and companies with high value are more likely to be favored by insurance institutions, who then take stakes in the company. This suggests that, to some extent, the shareholding ratio of insurance institutions and firm valuation are causally related, which could cause endogeneity issues in the model. Based on the two reasons above, a more reasonable model to evaluate the value selection function of insurance institutions is to utilize the t-period III's shareholding to regress the company value in t+1 period.

Based on the foregoing assumptions, we use the business value as the explained variable and the shareholding ratio of IIIs with a lag period as the explanatory variable to evaluate the first hypothesis of this study, namely, whether IIIs have the role of value selection. In this paper, the value selection model is created using the following linear regression model:

$$\text{ComValue}_{jit} = \beta_{j1} * \text{L.Shareh}_{jit} + \beta_{j2} * \text{Control}_{jit} + \alpha_{ji} + \vartheta_{jt} + \varepsilon_{jit} \tag{A}$$

where $ComValue_{jit}$ is the value of listed company i in period t; $Shareh_{jit}$ is the shareholding ratio of listed company i in period t; $Control_{jit}$ is the control variable; $\alpha_{ji}$ and $\vartheta_{jt}$ are the sample companies' individual and time effects, respectively; and $\varepsilon_{jit}$ is random error. Furthermore, j = 1, 2, with 1 denoting long-term stock holding IIIs, and 2 denoting FIFO IIIs. L. in model (A) represents the first-order lag operator, and the one-period lag is used because the proof of the value selection ability of IIIs requires a certain period of time to test, which has a lag; however, the endogeneity problem can be solved to a certain extent. The value selection model (A) is primarily utilized to see if the IIIs presented in Hypothesis 1 have a value selection function.

We use the change in company value (*D. ComValue_{jit}*) as the explanatory variable and the stock holdings of IIIs as the explained variable to test the second and third hypotheses of this paper, namely, whether IIIs have a value-creating function. The change in the shareholding ratio of insurance institutions (*D.Shareh_{jit}*) is employed as an explanatory variable in the value

creation model that follows (B):

$$D.\text{ComValue}_{jit} = \gamma_{j1} * D.\text{Shareh}_{jit} + \gamma_{j2} * \text{Control}_{jit} + \alpha_{ji} + \vartheta_{jt} + \varepsilon_{jit} \qquad (B)$$

What needs to be clarified in model (B) is the meaning of D., which stands for the first-order difference operator, especially:

$$D.ComValue_{jit} = ComValue_{jit} - ComValue_{jit-1}$$

$$D.Shareh_{jit} = Shareh_{jit} - Shareh_{jit-1}$$

Model (B) is utilised as a model for evaluating value creation because even if Model (A) is constructed, it can only determine the value selection function of IIIs; however, determining whether IIIs have a value creation function is challenging. Model (B) must establish the explained variable as a difference operator to observe the improvement of the company's value to test if the III's shareholding may help enhance the company's worth. Changes in the shareholding ratio of insurance institutions are also used as explanatory variables in model (B), primarily for two reasons: Changes in insurance institutions' shareholding ratios can reflect their preferences at a given time—buying or selling stocks is an important way for insurance institutions to participate in corporate governance and influence the value of companies, referred to as "voting with hands" and "voting with feet".

This paper determines the indicators to measure the company's value after establishing the models (A) and (B). Most studies, like those by Cornett et al. [51], Ferreira and Matos [52] and Healy [53], employ Tobin's Q [54] value to evaluate firm value, but China's stock market's price discovery function is in a nascent stage, and there is a strong speculative atmosphere. It is difficult to determine the true worth of publicly traded companies based on Tobin's Q [54]. Therefore, this article selects three measures to estimate the value of organisations after meticulous screening. Our first indicator, based on Chaganti and Damanpour [55] and Tang and Song [27], chooses earnings per share (EPS) as the key explanatory variable. Second, two alternative proxies, return on assets (ROA) [51, 56] and return on equity (ROE) [19, 57], are utilised as robustness test indicators. The parameters $\beta_{j1}$ and $\gamma_{j1}$ test the value selection and value creation functions of IIIs, respectively, where j as mentioned previously, takes 1 for long-term shareholding IIIs, and 2 for FIFO insurance institutional investment. The estimated parameters $\beta_{11}$ and $\beta_{21}$ are expected to be positive under assumption H1, and $\gamma_{21}$ is expected to be negative or insignificant under assumption H2, implying that FIFO IIIs do not have the function of value creation; the parameter $\gamma_{11}$ is expected to be positive under assumption H3a, but $\gamma_{11}$ is expected to be negative or insignificant in H3b.

## Selection of instrumental variables

There are two types of endogenous difficulties that frequently arise during the estimation of panel models: (1) It is difficult to quantify the effect of factors related to IIIs on the value of the held company, such as measurement errors and omitted variables; (2) The value of the held company and the holdings of IIIs could have a reverse causal relationship. This study addresses three aspects of the first type of endogenous problem: First, it improves the data and restores the data's legitimacy. This work does not employ databases to handle the problem of missing data, and instead combines the three major databases of WIND, CMSAR, and RESSET to collect complete sample data. Secondly, it selects an acceptable model estimation approach. The data in this research is balanced panel data, and we use the panel data to develop a fixed effect model. The fixed effect model can ignore the influencing factors of individuals that do not

change with time such as corporate culture, company creation date, corporate character, and other factors. Thirdly, it incorporates control variables. To limit the interference of residuals on the shareholding ratio, control factors such as the company's growth, capital scale, and so on are introduced to the model.

This paper uses the lagging one period of the shareholding ratio of insurance institutions as the explanatory variable for constructing the value selection model in the model design to mitigate the negative impact of company value on the shareholding ratio of IIIs in the second type of endogenous problem. Some researchers have utilised the turnover rate of tradable shares as an instrumental variable of institutional investors' ownership ratio in the past literature [27], however this indicator is not appropriate for the IIIs analysed in this article. Instrumental variables must meet two conditions: they must be strongly correlated with endogenous variables, and must not be correlated with model disturbances. The stock turnover rate does not match the first criteria, because it has a very weak link with the shareholding ratio of IIIs. The correlation coefficient between the overall turnover rate and the shareholding ratio is nearly 0, and the correlation coefficient between the outstanding share turnover rate and the shareholding ratio is just 0.012, which is not significant. Relevant data can be requested from the author. Therefore, alternative indicators for instrumental factors must be sought. This paper uses the institutional investor equity concentration of sample companies as a tool variable, that is, the proportion of the top three and top five institutional investors in the invested company (the specific equity concentration recorded as InsFir3 and InsFir5) as an instrumental variable, based on the relevant research results of Demsetz [58] and Demsetz and Lehn [59] in the field of corporate governance and referring to the idea of selecting instrumental variables by Hansen et al. [25]. Simultaneously, the higher-order lag term (L3.EPS and L4.EPS) of the insurance company's shareholding ratio is used as an instrumental variable to mitigate the problem of weak instrumental variables [60].

The specific equity concentration variables InsFir3 and InsFir5 are employed as the instrument variables of the III's shareholding ratio in this research to investigate the hypothesis that "the ownership structure has nothing to do with the company's success". According to some academics, the ownership structure will have an impact on the company's performance because of the principal-agent problem generated by the separation of the two sets of rights [61–64].

Demsetz [58], on the other hand, questioned the above-mentioned viewpoints, and asserted that the company's shareholding structure is the game's equilibrium conclusion after each player analyses the advantages and negatives to maximize their personal interests. Thus, the company's overall ownership structure has little impact on its success. Although the separation of control and ownership can cause principal-agent difficulties [63], it can also aid in increasing the quantity of the funds obtained by extending the company's scale, and improve the company's operating efficiency by increasing the degree of professional specialization (i.e., separation of capital providers from professional agents). The advantages of separating control and ownership are counterbalanced by the potential risks. Therefore, the equity structure has no bearing on the company's performance as otherwise, there would not be a significant variety in the equity arrangements of companies in the long run [25].

The rationale for the instrumental variables in this paper is as follows: First, the fixed effect model controls the disturbance that a company's individual heterogeneity has on the individual heterogeneity of the stock structure of institutional investors within the holding business by partially excluding the company's individual effect. Second, even if there are endogenous concerns in institutional investor ownership concentration (Insfir3, Insfir5) in the instrumental variable sample firm [59], the ownership structure is mostly influenced by the company size, profitability, and industry. Including control variables, such as firm size, improves the

usefulness of instrumental variables to a certain degree in the model. Third, using ownership concentration degree as a control variable further reduces the mutual disturbance between residuals from the model and instrumental variables by controlling for important information from institutional investors who are also significant shareholders. Fourth, the number of instrumental variables outnumbers the number of explanatory variables that are endogenous as over-identification is more successful in large samples [65]. The validity of instrumental variables can be verified using methods such as the Sargan test, which also happens to identify a test method that does not yet exist in the case of over-identification. Fifth, meet the necessary instrumental variable test standards, such as the identification test, weak instrumental variable test, and so on.

## Model design for counterfactual testing

This paper creates a counterfactual test model to check the reliability of the model conclusions to examine whether the model's inherent flaws are effectively controlled. Theoretically, IIIs' future shareholding should have no impact on the existing holding company's performance. When the parameters $\beta_{j1}$ and $\gamma_{j1}$ to be estimated are significant in the measurement model constructed by the future shareholding ratio, the information about the insurance company's equity structure indicators and the held company's performance is still in the model's disturbance terms, and as an instrumental variable, it does not effectively solve the model's endogeneity. Similarly, when they are not significant, the endogenous problem in the model is effectively controlled to some extent. The following counterfactual testing models (C) and (D) are based on this concept:

$$L2.ComValue_{jit} = \beta_{j1} * L.Shareh_{jit} + \beta_{j2} * L2.Control_{jit} + \alpha_{ji} + \vartheta_{jt} + \varepsilon_{jit} \tag{C}$$

$$L2.(D.ComValue_{jit}) = \gamma_{j1} * D.Shareh_{jit} + \gamma_{j2} * L2.Control_{jit} + \alpha_{ji} + \vartheta_{jt} + \varepsilon_{jit} \tag{D}$$

L2. in (C) and (D) represents the second-order lag operator, which lags the explanatory variable by two periods. Except for the shareholding ratio of IIIs and their shareholding changes, which are the same as in models (A) and (B), the rest of the control variables are lagged by two periods to verify the relationship between future and current shareholding, and to conduct repeated counterfactual tests of the relationship between performances in formulas (C) and (D).

# Data

## Data processing

**Sample period.** The sample period for this study is from the first quarter of 2012 to the first quarter of 2018, and the research objects for empirical research are A-share listed companies on the Shanghai and Shenzhen stock markets in China. The data is collected on a quarterly basis, with a total of 25 quarters. Furthermore, while calculating the holding duration of IIIs, the first quarter of 2010 is used as the starting point, implying that the computation of the holding term of IIIs begins in the first quarter of 2010. Despite Chinese insurance funds having entered the capital market long before the subprime mortgage crisis, this paper uses 2010 as the starting point for the calculations because the Insurance Law empowering insurance companies to make equity investments was promulgated in 2009. According to statistics on publicly traded companies, insurance institutions owned less than 200 publicly traded companies in 2009, with a low shareholding ratio. Thus, this year is excluded, and the calculations begin in 2010 as the effects of ownership was insignificant in 2009.

The data shows that as of the end of the first quarter of 2018, there were less than ten companies where insurance companies held shares for periods of more than 20 quarters (5 years). This indicates that our calculation of shareholding periods since 2010 essentially covers the shareholding cycle of all companies held by insured institutions, and validates the starting point and the sample's integrity.

The period between 2010 and 2011 is not included in the sample as the holdings by insurance companies were not substantial during this period and had a low shareholding ratio and a small scale [27]. Thus, the time period chosen is from the first quarter of 2012 up till the first quarter of 2018. China's capital market suffered stock market volatility in 2015. The "Vanke-Baoneng Battle" has raised concerns about insurance capital, as has the issue of employing short-term insurance funds for long-term investment by insurance capital. In July 2015, the Baoneng Group's Qianhai Life Insurance Company purchased Vanke shares on the secondary market, amounting for 5% of the company's total share capital. Baoneng, working with Qianhai Life Insurance, Jushenghua, and other parties through multi-party activities, exceeded China Resources Group, Vanke's original largest shareholder, for the first time in less than a month, and suggested calling an extraordinary general meeting to recall all directors. At the same time, Vanke used a private placement to introduce its shareholder Shenzhen Metro Group to pay the consideration. The probe has included the China Securities Regulatory Commission and the Shenzhen Stock Exchange. Shenzhen Metro Group eventually became the largest shareholder, Baoneng gradually withdrew from the competition, the former chairman of Vanke stepped down, and the war between Vanke and Baoneng came to an end. The sample in this research spans the three-year sample period before and after 2015, yielding a relatively complete capital market cycle. A balanced panel data sample with the data format "company-quarter" covering a total of 25 quarters for this study.

**Data resource.** The model's economic indicator data originates from three primary databases: WIND, CMSAR, and RESSET, as well as some companies' quarterly financial reports. The RESSET database is primarily used to calculate financial metrics such as company growth, capital structure, and scale. At the same time, missing data is filled using WIND, CMSAR databases, and quarterly financial reports. If a sample's missing observation data is greater than 3/5 of the sample time zone's length, the sample will be removed immediately. The limits will be eased as far as practicable if the company has insurance institutional investors holding shares. Interpolation (average of the upper and lower periods of missing data) and observation were used to fill up the gaps. Because the number of listed firms held by insurance companies in the early stages is quite limited, and missing data on the shareholding ratio is particularly serious, the goal is to keep as large a sample of listed companies held by insurance institutional investors as possible. The data of the shareholding ratio of IIIs originates from CMSAR database. The missing pieces are primarily from databases such as WIND. Data on the shareholding ratio of some insurance institutional investors is not disclosed at the end of the quarter, but rather at various intervals throughout the year. This study uses sample observations from the most recent release before the conclusion of the quarter to assure the data's richness. Other non-financial variables are considered in the same way as their end-of-quarter proxy value. Between the WIND database and the other two databases, there is a significant gap. This article is only intended to fill in the missing data for reference purposes, in order to ensure the data's richness and accuracy.

**Data processing.** This paper excludes abnormal samples such as "ST" and "*ST" shares, as well as the sample of financial companies, which yielded 1,654 samples. If a listed company's financial state or other conditions are anomalous in China's capital market, the Shanghai and Shenzhen Stock Exchanges will give the listed company's shares special treatment. This form of stock is known as ST stock because of the "special treatment" it receives. If a stock's name is

prefixed with ST, it carries investment risk; if *ST is added, it has the danger of being delisted. The firm samples are then separated into two categories based on their investment horizon: long-term shareholding type and short-term shareholding type, with the latter being referred to as FIFO type in this article. The III shareholding period indicator (InsDura) is calculated as follows: From 2020 onwards, we will calculate whether a publicly traded firm is owned by insurance institutions. If the frequency is based on quarter, there are a total of 33 intervals of quarterly period $t_j$ from the first quarter of 2010 to the end of the first quarter of 2018. If the stock of listed company i is held by IIIs in a specified interval $t_j$ (j = 1, 2. . ., 33), the variable (yes_noIns) associated with $t_j$ will be marked 1. Conversely, if no insurance institution holds shares of the listed company i in a given interval $t_j$, the variable (yes_noIns) corresponding to the listed company i in that interval $t_j$ is set to 0; then the interval in which 1 appears in the sample individual i consecutively is set to $m_{ik}$ (where k = 1,2. . .,n, where n is the maximum number of consecutive intervals), and then the consecutive terms $m_{ik}$ are added together, and recorded as $t_{ij}$. As a result, the holding duration of IIIs is as follows:

$$InsDuraf(i) = \begin{cases} t_{ij}, & \text{If the sample individual i in the interval is held by IIIs;} \\ 0, & \text{elsewhere} \end{cases}$$

The average holding time of the selected sample by insurance firms is 6.9 quarters, according to the aforementioned method. Elyasiani and Jia [66] divided the long-term and FIFO samples based on whether it was greater than the index mean. This work addresses the symmetry of the long and short multiples to assure the sample's adequacy while also distinguishing the structural characteristics of the holding duration. IIIs with more than 1.25 times the average holding time (greater than 9 quarters) are considered long-term holding types in this paper, while those with less than 0.75 times (less than 6 quarters) are considered FIFO holding types. The ratio is set at 1.5 times and 0.5 times, respectively, in the robustness test, i.e., more than 11 quarters are recorded as longer-term holdings, and less than 4 quarters are recorded as shorter-term holdings.

**Descriptive statistics.** The explained variable is the firm value (expressed by EPS, ROA, and ROE), and according to Cornett et al. [51], the key explanatory variable is the shareholding ratio of IIIs. Table 1 shows the detailed definitions. The control variables in the model include: (1) Corporate governance structure (Conc), which is measured by the company's ownership concentration. According to previous research literature, firm value does not alter ownership structure, and hence there is no reverse causal relationship between ownership structure and company value [27, 67–69]. The mutual interference between the model residual and the ownership structure can be moderately reduced if the control variable includes the ownership structure index, which improves the effectiveness of the institutional investor ownership concentration (Insfir3, Insfir5) as an instrumental variable. (2) Company growth (Grow), following Aggarwal et al. [29], this work controls for the company's growth as it may have an impact on its value. (3) Financial leverage (AssLia). An asset-liability ratio is selected to measure the impact of financial leverage as financial leverage has an influence on firm value [70, 71]. (4) company size (Size). The impact of company size is factored into the model as a great number of studies have shown that company size has an impact on the company's value. (5) Institutional investor heterogeneity (NonIns). The characteristics of shareholders impact firm value, according to Huang et al. [54], Tang and Li [71], Song et al. [72] and Li et al. [73], thus, this article addresses the impact of non-IIIs. (6) The lag term (PreValue) of the company's value measurement variable is also measured as it may be more difficult to enhance the performance of listed firms with high value. Thus, the previous period's company value is introduced as a control variable to models (A) and (B) to mitigate the endogenous problem [27, 41]. Higher-

**Table 1. Variables definitions.**

| Index sign | | Index meaning | Index name and calculation method |
|---|---|---|---|
| Explained variables | EPS | The proxy variable to measure the value of a company | $Eearnings\ per\ share = \dfrac{Current\ net\ profit\ attributable\ to\ common\ shareholders}{\sum(Number\ of\ outstanding\ common\ shares*portion\ of\ the\ reporting\ period\ those\ shares\ covered)/12}$ |
| | ROA | | $Return\ on\ Assets = \dfrac{Net\ Income\ (\ including\ minority\ interests)*2}{initial\ total\ assets+ending\ total\ assets} * 100\%$ |
| | ROE | | $Return\ on\ equity = \dfrac{Net\ profit\ attributable\ to\ parent\ company\ shareholders*2}{Initial\ equity\ attributable\ to\ shareholders\ of\ the\ parent\ company\ +\ ending\ equity\ attributable\ to\ shareholders\ of\ the\ parent\ company}$ |
| Explanatory variable | Shareh | Shareholding ratio of IIIs | The proportion of institutional investors (insurance companies) holding outstanding shares of listed businesses, according to data from the Wind, RESET, and CMSAR databases. |
| Control variable | Grow | Company growth | $Growth\ rate\ earnings\ per\ share = \dfrac{earnings\ per\ share\ of\ this\ quater - earnings\ per\ share\ of\ prior\ quater}{earnings\ per\ share\ of\ prior\ quarter} * 100\%$ |
| | AssLia | Capital Structure Financial Leverage | Asset-liability ratio = Total liability/total assets |
| | Size | Company scale | Logarithmic total assets = natural logarithm of total assets |
| | Conc | Corporate governance structure | Ownership concentration = The total shareholding ratio of the top three shareholders |
| | NonIns | Control institutional investors' heterogenous difference | Shareholding ratio of non-insurance companies = sum of shareholding ratios of non-insurance investors including four major institutional investors, namely, funds, security financial products, banks and pension funds. |
| | PreValue | Company value in the previous period | The company value of the listed company in the previous period is L.ComValue in model (A) and L2. ComValue in model (B). PreValue corresponds to three indicators of company value: EPS, ROA, ROE |

order lag terms (L3.ComValue and L4.ComValue) are likewise considered as instrumental variables [41, 60].

All variables are winsorized by 1 percent and 99 percent in this work to eliminate the influence of outliers on the model. Table 2 presents summary statistics for the variables. Panel A shows that long-term and FIFO III shareholding ratios are roughly 1.7417% and 0.2414%, respectively, with long-term investors having the greatest shareholding ratio of 58.820%. FIFO investors had the highest shareholding ratio at 21.820%. The ratio of the total shareholding ratio of insurance institutional investors to the total number of quarters is used to calculate the average shareholding ratio of insurance institutional investors. If the quarters with a zero-shareholding percentage are omitted, the overall sample's average shareholding ratio is around 2.5 percent. In addition, Panel B shows the shareholding ratio and shareholding term of insurance institutions under various company values after grouping company value EPS according to the average value. The mean shareholding ratio and holding term of insurance institutions are significantly higher in the group with higher company value (EPS> = median) than in the group with lower business value (EPS<median). It is first demonstrated that IIIs have a value selection function.

## Model regression analysis and counterfactual testing

**Correlation test.** In this paper, each variable is correlated, and the correlation coefficient matrix is provided in Table 3. The correlation between the major variables is poor except for the three types of corporate performance indicators, which includes EPS, ROA, and ROE. The remaining variables' correlation coefficients are almost all below 0.5, indicating that the model does not have severe multicollinearity issues. The correlation coefficients between the insurance shareholding ratio and the company value index are 0.122, 0.037, and 0.055, respectively, indicating a positive association between the insurance shareholding ratio and the company value. Simultaneously, economic variables such as company value measurement indicators,

**Table 2. Summary statistics.**

| Panel A | | | Long-term holding | | | | | | | | |
|---|---|---|---|---|---|---|---|---|---|---|---|
| Index | EPS | *ROA* | *ROE* | Shareh | Insfir3 | Insfir5 | Grow | AssLia | *Size* | Conc | NonIns |
| Number of samples | 8700 | 8700 | 8700 | 8700 | 8700 | 8700 | 8700 | 8700 | 8700 | 8700 | 8700 |
| Mean | 0.4671 | 4.6927 | 5.3377 | 1.7417 | 0.8857 | 1.3346 | 15.8804 | 47.4281 | 2.0051 | 49.8864 | 6.2176 |
| Standard deviation | 0.5216 | 5.3864 | 5.8877 | 4.077 | 3.2786 | 3.7656 | 96.4969 | 19.5290 | 0.6158 | 16.6336 | 7.8271 |
| Minimum | -0.8344 | -29.4725 | -10.54 | 0 | 0 | 0 | -235.300 | 7.7649 | 0.2612 | 8.4986 | 0 |
| Median | 0.3553 | 3.8896 | 4.08 | 0.33 | 0 | 0 | 7.5449 | 47.0514 | 1.9206 | 50.2304 | 3.585 |
| Maximum | 2.2213 | 46.3117 | 21.4900 | 58.82 | 36.3737 | 46.8682 | 324.8260 | 86.358 | 4.3885 | 98.3571 | 102.762 |
| | | | Fast in and fast out | | | | | | | | |
| Number of samples | 19500 | 19500 | 19500 | 19500 | 19500 | 19500 | 19500 | 19500 | 19500 | 19500 | 19500 |
| Mean | 0.2851 | 3.7367 | 4.1145 | 0.2414 | 0.0617 | 0.1215 | 18.1226 | 42.9522 | 1.6296 | 47.5812 | 4.9567 |
| Standard deviation | 0.4033 | 5.8575 | 5.784 | 0.8611 | 0.5594 | 0.6989 | 114.5397 | 21.5781 | 0.5117 | 15.18 | 7.7091 |
| Minimum | -0.8344 | -29.4725 | -10.54 | 0 | 0 | 0 | -235.294 | 3.8217 | -0.598 | 0.5648 | 0 |
| Median | 0.2157 | 3.2157 | 3.05 | 0 | 0 | 0 | 5.1728 | 41.8841 | 1.5647 | 46.9198 | 1.9931 |
| Maximum | 2.2213 | 46.3117 | 21.49 | 21.82 | 18.75 | 18.75 | 324.826 | 92.1799 | 3.4195 | 97.53 | 96.6152 |
| Panel B | EPS> = median | | | EPS<median | | Difference | | | | | |
| | | | Long-term holding | | | | | | | | |
| | Mean | | | Mean | | Mean | | | | | |
| Shareholding ratio | 2.0480 | | | 1.4864 | | 0.5616*** | | | | | |
| Holding period | 4.5037 | | | 3.1912 | | 1.3125*** | | | | | |
| | | | Fast in and fast out | | | | | | | | |
| Shareholding ratio | 0. 2792 | | | 0. 1858 | | 0.0934*** | | | | | |
| Holding period | 0.3247 | | | 0.2364 | | 0.0883*** | | | | | |

shareholding ratio of IIIs, and shareholding concentration all reject the existence of unit root at the 1% or 5% significant level, based on the panel unit root tests such as the Levin-Lin-Chu (LLC) test [74].

## Model regression analysis

The data generated in this paper is strongly balanced panel data (firm-year panel). We adopt a fixed effect model for model regression following Aggarwal et al. [29] and Massa et al. [75]. This is for a number of reasons: First, from an economic standpoint, omitted variables, such as corporate culture, executive characteristics, and so on, are often difficult to quantify, and "heterogeneous" differences in companies, which are difficult to observe and measure, frequently lead to serious endogenous problems. The fixed-effect model can lessen the endogenous problem produced by the bias interference of omitted variables as the differential process in the estimated approach of the fixed-effect model can eliminate "time-invariant" firm heterogeneity to some extent. Second, this paper used statistical analysis such as the F test and the Hausman test from a statistical standpoint. The results reveal that at a 1% significance level, the fixed effect model is the most appropriate. The author may provide F test data upon request, and the values of Hausman test statistics can be found in Table 4. The Two-Stage Least Squares (2SLS) estimation results are used in this paper for the long-term holding type and the fast-in-fast-out type; simultaneously, the Limited Information Maximum Likelihood estimation (LIML) method, which is insensitive to weak instrumental variables, is used to obtain the model parameters. The reasonableness of the instrumental variable selection is indirectly validated if the difference between the two estimations is minimal. Thirdly, we diagnose the model

**Table 3. Pearson correlation coefficient.**

| Index | EPS | ROA | ROE | Shareh | Grow | AssLia | Size | Conc | NonIns |
|---|---|---|---|---|---|---|---|---|---|
| **EPS** | 1 | | | | | | | | |
| **ROA** | 0.711*** | 1 | | | | | | | |
| **ROE** | 0.594*** | 0.567*** | 1 | | | | | | |
| **Shareh** | 0.162*** | 0.067*** | 0.087*** | 1 | | | | | |
| **Grow** | 0.197*** | 0.200*** | 0.342*** | 0.007 | 1 | | | | |
| **AssLia** | -0.095*** | -0.326*** | -0.084*** | 0.011* | 0.011* | 1 | | | |
| **Size** | 0.224*** | -0.064*** | 0.128*** | 0.119*** | 0.049*** | 0.518*** | 1 | | |
| **Conc** | 0.161*** | 0.094*** | 0.126*** | -0.033*** | 0.003 | 0.040*** | 0.276*** | 1 | |
| **NonIns** | 0.278*** | 0.255*** | 0.265*** | 0.086*** | 0.059*** | -0.127*** | -0.049*** | -0.035*** | 1 |

Note: Table 3 preserves three decimal places since the correlation coefficient has only three decimal places.

residuals and draw a scatter plot of residuals and model fitting values at the same time, demonstrating that the model has no heteroscedasticity problem. Similarly, since the Generalized Method of Moments (GMM) regression method is suitable for the case where the model has heteroscedasticity, the GMM estimation method is not used in this paper. The regression results of the GMM method are available to the readers upon request. The regression findings are as follows: the value selection model results correspond to model (A), and hypothesis 1 is tested, as shown in Table 4. The value creation model's outcomes correlate to model (B), as shown in Table 5. Columns (3) and (4) of Table 5 tests hypothesis 2 and determines whether FIFO IIIs have a value creation function. The results in columns (1) and (2) test hypothesis 3 and determine whether long-term shareholding insurance institutions are value creators or value suppressors.

The numbers in this paper's table are rounded to four decimal places, however Table 4 is rounded to five decimal places because the LIML and 2SLS findings only differ in the fifth decimal place. Tables 4 and 5 show that the instrumental variables chosen by the model are statistically appropriate. Concerning the problem of weak instrumental variables, LIML's conclusions in the two tables are essentially identical to the 2SLS estimation results, and the CD value and KP value of the weak instrumental variable test are both greater than the critical value of 10, indicating that the model has no obvious weak instrumental variable problem. The Hensen J statistic in the two tables did not reject the null hypothesis of "instrumental variables are exogenous" in the over-identification test, showing that instrumental variable selection is statistically sensible. Furthermore, the Hausman statistic demonstrates that the fixed-effect model is superior to the random-effect model, validating the rationale of using fixed effects in this paper.

The findings of the value selection model (A) are shown in Table 4. The shareholding ratio coefficient $\beta_{11}$ (0.00463) of insurance institutions was significantly positive at the level of 1% for long-term shareholders in the preceding period, as shown in the table, and is favourably associated to the company value. $\beta_{21}$ (0.00509) is likewise significantly positive at the 10% level for FIFO IIIs. Table 4 shows that the research hypothesis H1 is correct, demonstrating that insurance companies have a value selection function. Long-term shareholding insurance companies are willing to pay a cost to select high-quality companies to generate consistent equity returns. Conversely, FIFO insurance companies will actively use their information collection advantages to find high-quality companies to hold shares.

Table 5 shows the empirical results of the value creation model (B), in which columns (3) and (4) show that FIFO insurance institutions do not have a value creation function, and their

**Table 4. Value selection model (the explained variable is earnings per share:EPS).**

| Variable | Long-term holding | | Fast in and fast out | |
|---|---|---|---|---|
| | IV (LIML) (1) | IV (2SLS) (2) | IV (LIML) (3) | IV (2SLS) (4) |
| L. Shareh | **0.00463**\*\*\* | **0.00463**\*\*\* | **0.00509**\* | **0.00509**\* |
| | **(0.00145)** | **(0.00145)** | **(0.00278)** | **(0.00278)** |
| L.EPS | 0.56547\*\*\* | 0.56546\*\*\* | 0.52198\*\*\* | 0.52199\*\*\* |
| | (0.04151) | (0.04150) | (0.04915) | (0.04914) |
| Grow | 0.00068\*\*\* | 0.00068\*\*\* | 0.00065\*\*\* | 0.00065\*\*\* |
| | (0.00003) | (0.00003) | (0.00002) | (0.00002) |
| AssLia | -0.00020 | -0.00020 | -0.00057\* | -0.00057\* |
| | (0.00039) | (0.00039) | (0.00034) | (0.00034) |
| LogAss | -0.06924\*\* | -0.06924\*\* | 0.02408 | 0.02408 |
| | (0.03287) | (0.03287) | (0.01920) | (0.01920) |
| Conc | 0.00117\* | 0.00117\* | 0.00014 | 0.00014 |
| | (0.00061) | (0.00061) | (0.00040) | (0.00040) |
| NonIns | 0.00186\*\*\* | 0.00186\*\*\* | 0.00230\*\*\* | 0.00230\*\*\* |
| | (0.00043) | (0.00043) | (0.00038) | (0.00038) |
| Industry variables | Controlled | Controlled | controlled | controlled |
| Joint significance | 40.96\*\*\* | 40.96\*\*\* | 118.71\*\*\* | 118.72\*\*\* |
| Under-identification | 1027.685\*\*\* | 1027.685\*\*\* | 175.280\*\*\* | 175.280\*\*\* |
| Weak instrumental variables | 1681.579/ 199.382 | 1681.579/ 199.382 | 3121.977/ 38.021 | 3121.977/ 38.021 |
| Hansen J | 3.054 | 3.054 | 3.652 | 3.652 |
| Hausman | | 199.49\*\*\* | | 2459.68\*\*\* |
| Number of Observations N | 8715 | 8715 | 19500 | 19500 |

Note: The joint significance of the time dummy variables is referred to as joint significance. The symbols "*," "**," and "***" in the table indicate that the statistic is significant at the 10%, 5%, and 1% levels, respectively. "In the first-stage regression, the instrumental variables utilized have no explanatory power for endogenous variables," is the null hypothesis of the under-identification test, and the value of the under-identification test in this work considerably rejects the null hypothesis. The Cragg-Donald Wald (CD) statistic and the Kleibergen-Paap (KP) statistic are both weak instrumental variable tests. Stock et al [76] argue that even if instrumental variables pass the under-identified test, they may still exhibit "substantial distortion" (size distortion). Hence the Stock-Yogo statistic is based on this assumption. It is more difficult to reject the null hypothesis—"the instrumental variable is strongly correlated with the endogenous variable"—when the KP value is greater than the Stock-Yogo statistic value. Since the KP value in this paper is greater than this value, it can be ascertained that there is no weak instrumental variable problem. The results of the over-identification test were reported by Hansen J. The number of instrumental factors in this study exceeds the number of endogenous variables, meeting the over-identification test's assumption. The results of Hansen J are not significant, and the basic hypothesis that "all instrumental factors are exogenous" is not rejected, which also reveal the logic behind the selection of model instrumental variables. This also applies to Table 5.

insurance institution shareholding ratio coefficient $\gamma_{21}$ (-0.0081, - 0.0080) is negative at the 10% significance level, indicating that the impact of changes in shareholdings by insurance institutions on company value is negative, confirming the proposed hypothesis H2. The FIFO III's shareholding has no positive correlation with the company's value movements and serves no value creation role.

The results of long-term shareholding insurance institutions are reported in columns (1) and (2) of Table 5. The long-term investor shareholding ratio coefficient $\gamma_{11}$ (0.0168, 0.0167) is positive at the 5% significance level, as shown in the table. As a result, hypothesis H3a in the competitive hypothesis H3 is confirmed, i.e., long-term equity-holding IIIs have a value

**Table 5. Value creation model (the explained variable is the change in earnings per share, namely: D.EPS).**

| Variable | Long-term holding | | Fast in and fast out | |
|---|---|---|---|---|
| | IV (LIML) (1) | IV (2SLS) (2) | IV (LIML) (3) | IV (2SLS) (4) |
| D.Shareh | 0.0168** | 0.0167** | -0.0081* | -0.0080* |
| | (0.008) | (0.008) | (0.005) | (0.005) |
| L2.EPS | -0.1920*** | -0.1919*** | -0.2310*** | -0.2310*** |
| | (0.016) | (0.016) | (0.014) | (0.014) |
| Grow | 0.0006*** | 0.0006*** | 0.0005*** | 0.0005*** |
| | (0.000) | (0.000) | (0.000) | (0.000) |
| AssLia | 0.0003 | 0.0003 | -0.0003 | -0.0003 |
| | (0.000) | (0.000) | (0.000) | (0.000) |
| LogAss | -0.0659** | -0.0659** | -0.0170 | -0.0170 |
| | (0.031) | (0.031) | (0.015) | (0.015) |
| Conc | -0.0004 | -0.0004 | -0.0002 | -0.0002 |
| | (0.001) | (0.001) | (0.000) | (0.000) |
| NonIns | 0.0008** | 0.0008** | 0.0008*** | 0.0008*** |
| | (0.000) | (0.000) | (0.000) | (0.000) |
| Industry variables | controlled | Controlled | controlled | controlled |
| Joint significance | 28.60*** | 28.61*** | 49.38*** | 49.38*** |
| Under-identification | 86.749*** | 86.749*** | 219.641*** | 219.641*** |
| Weak instrumental variables | 56.126 / 20.382 | 56.126 / 20.382 | 455.985 / 55.634 | 455.985 / 55.634 |
| Hansen J | 0.282 | 0.282 | 3.288 | 3.288 |
| Hausman | | 693.99*** | | 2548.53*** |
| Number of samples N | 8715 | 8715 | 19500 | 19500 |

Note: The joint significance of the time dummy variables is referred to as joint significance. The symbols "*," "**," and "***" in the table indicate that the statistic is significant at the 10%, 5%, and 1% levels, respectively. "In the first-stage regression, the instrumental variables utilized have no explanatory power for endogenous variables," is the null hypothesis of the under-identification test, and the value of the under-identification test in this work considerably rejects the null hypothesis. The Cragg-Donald Wald (CD) statistic and the Kleibergen-Paap (KP) statistic are both weak instrumental variable tests. Stock et al [76] argue that even if instrumental variables pass the under-identified test, they may still exhibit "substantial distortion" (size distortion). Hence the Stock-Yogo statistic is based on this assumption. It is more difficult to reject the null hypothesis—"the instrumental variable is strongly correlated with the endogenous variable"—when the KP value is greater than the Stock-Yogo statistic value. Since the KP value in this paper is greater than this value, it can be ascertained that there is no weak instrumental variable problem. The results of the over-identification test were reported by Hansen J. The number of instrumental factors in this study exceeds the number of endogenous variables, meeting the over-identification test's assumption. The results of Hansen J are not significant, and the basic hypothesis that "all instrumental factors are exogenous" is not rejected, which also reveal the logic behind the selection of model instrumental variables

creation function. The main logical premise is that presently Chinese insurance funds have high stability and low fund recovery pressure, and long-term shareholding by insurance funds fully exploits their features and advantages while also promoting the value appreciation of the held company. However, due to its low shareholding ratio and unwillingness or inability to bear the cost of governance, the FIFO type is less involved in corporate governance and instead uses the advantages of information collection to select high-quality companies with high internal governance transparency. Some FIFO IIIs even demonstrated the phenomenon of "investors-retail," which is very similar to the investment goals of short-sighted managers. Conspiracy to boost short-term returns, trigger group speculation, swiftly drive up the stock

price and cash out, all of which have a negative influence on value creation and adversely affects the company's long-term development strategy, is simple to execute. As a result, long-term IIIs have the function of value creation, whereas FIFO IIIs do not.

## Counterfactual test

We built models (C) and (D) for counterfactual testing in the preceding discussion [77, 78]. Theoretically, future III shareholding will have no impact on the current holding company's value, i.e., the shareholding ratio of insurance institutions in period t+1 has no impact on the firm's worth in period t. The explained variables and key control variables, other than the shareholding ratio of insurance institutions, are lagged two periods in the specified regression model, and the counterfactual test models (C) and (D) were constructed. Table 6 displays the model's output. The regression findings of the LIML and 2SLS approaches do not differ significantly, as shown in Tables 4 and 5. Thus, we only employ the LIML approach for regression in the following models. Table 6 shows that the parameters $\beta_{j1}$ (0.0008, 0.0011) estimated in columns (1) and (2), and $\gamma_{j1}$ (0.0023, 0.0048) estimated in columns (3) and (4) did not show

**Table 6. Counterfactual test model parameter estimation results.**

| Variable | Value selection model | | Value creation model | |
|---|---|---|---|---|
| | Explained variable: EPS (lagged by two periods) | | Explained variable: (D.EPS) (lagged by two periods) | |
| | Long-term holding (1) | Fast in and fast out (2) | Long-term holding (3) | Fast in and fast out (4) |
| L.Shareh | 0.0008 | 0.0011 | | |
| | (0.001) | (0.002) | | |
| D.Shareh | | | 0.0023 | 0.0048 |
| | | | (0.003) | (0.004) |
| L3.EPS | 0.7891*** | 0.7018*** | -0.2105*** | -0.2982*** |
| | (0.015) | (0.015) | (0.015) | (0.015) |
| L2.Grow | 0.0003*** | 0.0002*** | 0.0003*** | 0.0002*** |
| | (0.000) | (0.000) | (0.000) | (0.000) |
| L2.AssLia | 0.0007** | -0.0003 | 0.0006** | -0.0003* |
| | (0.000) | (0.000) | (0.000) | (0.000) |
| L2.LogAss | -0.0971*** | -0.0317** | -0.0949*** | -0.0312** |
| | (0.031) | (0.014) | (0.030) | (0.014) |
| L2.Conc | -0.0001 | -0.0001 | -0.0001 | -0.0001 |
| | (0.001) | (0.000) | (0.001) | (0.000) |
| L2.NonIns | 0.0017*** | 0.0019*** | 0.0016*** | 0.0020*** |
| | (0.000) | (0.000) | (0.000) | (0.000) |
| Industry variables | Controlled | Controlled | controlled | controlled |
| Number of Samples N | 8300 | 19500 | 8300 | 19500 |

Note: The table employs the LIML regression approach. The symbols "*," "**," and "***" in the table indicate that the statistic is significant at the 10%, 5%, and 1% levels, respectively. The explanatory variable is the shareholding ratio of an insurance institution with a first lag, and the explained variable is the two-period lagged EPS of the company. With a unified second lag, the control variable is the same as the explained variable. For weak instrumental variables in the table, the CD statistic and the KP statistic are both more than 10, and there is no weak instrumental variable. Table 6 does not give other statistics since the setup of the counterfactual test model is inconsistent with the actual situation. It primarily tests the relationship between the holdings of insurance institutions and the company value in the past.

significance. That is, there is no significant relationship between the shareholding ratio of insurance institutions and the company's value in the counterfactual value selection model (C) and the counterfactual value creation model (D). These results illustrate that the shareholding ratio of insurance institutions and the corresponding value selection model (A), and the disturbance terms in the value creation model (B) are relatively independent, and that the instrumental variable "institutional investor equity concentration" chosen in this paper can help address model endogeneity more effectively.

## Robustness test

This paper replaces the EPS with two alternative proxies: ROA and ROE, respectively, in the value selection model (A) and value creation model (B). The model parameters were re-estimated individually to assess the importance of the coefficients of the key examined variables. Table 7 displays the model's output. Long-term shareholding insurance institutions' value selection and creation are reported in columns (1)—(4) of Table 7, while FIFO insurance institutions' value selection and creation are reported in columns (5)—(8). It can be inferred from the table that the value selection model of the impact of long-term shareholding insurance institutions on listed companies, the coefficients of insurance shareholding ratio (0.0460, 0.0561), and the coefficients of insurance shareholding ratio (0.1575, 0.1767) in the sample of FIFO shareholding insurance institutions are positive at a significance level of more than 10%. This is consistent with the assumption that insurance institutions have the function of value selection in H1, and the results when EPS is the explained variable (Table 4). Among the results of the value creation model, the long-term shareholding value creation results are shown in columns (3) and (4). The coefficients of the insurance shareholding ratio (0.4833, 0.5683) are positive at the 10% significance level, which confirms hypothesis H3a. The FIFO shareholding value creation results are shown in columns (7) and (8). At the 5% significance level, the coefficients of the insurance shareholding ratio (-0.6118, -0.6232) are negative, which is consistent with the theoretical analysis of hypothesis H2. The results of the value creation model are also consistent with Table 5.

## Further research

The sample type—categorization criteria, is further narrowed in this research to explore the performance of value selection and creation methods across longer and shorter holding periods. This classification is a more in-depth research challenge and a strong evaluation of the results of this work from a different perspective. The comparison also reflects the stability of the value creation and value selection functions. The average holding period of insurance institutions above 1.5 times is recorded as a sub-sample with a longer holding period (long-term holding type 2). Similarly, holding periods of less than 0.5 times are recorded as a subsample of FIFO holdings with more frequent investment conversions (FIFO type 2), and model parameters are estimated using the value-selection model (A) and value-selection model (B). Table 8 displays the model's output. The FIFO type 2 IIIs have a more obvious value selection function than the FIFO type IIIs in Table 4, as shown in Table 8. The coefficient increased from 0.00509 (column (3) of Table 4) to 0.0170 (column (2) of Table 8), and the significance level increased from 10% to 5%, indicating that IIIs with shorter-term investment horizons face greater pressure to maintain the value of debt funds, and will actively utilize its advantages in information collection to seek high-quality companies to maximize the value of their debt funds.

Long-term holding type 2 has a substantial value selection function with a slight improvement in the coefficient (0.0048 (column (1) of Table 8) > 0.00463 (column (1) of Table 4), indicating that insurers are more cautious for longer-term holdings. The value creation effect

**Table 7. Test of alternative proxies.**

| Variable | Long-term holding | | | | Fast in and fast out | | | |
|---|---|---|---|---|---|---|---|---|
| | ROA | ROE | D. ROA | D. ROE | ROA | ROE | D. ROA | D. ROE |
| | Value selection model | | Value creation model | | Value selection model | | Value creation model | |
| | (1) | (2) | (3) | (4) | (5) | (6) | (7) | (8) |
| L.Shareh | **0.0460**\*\* | **0.0561**\*\* | | | **0.1575**\* | **0.1767**\*\*\* | | |
| | **(0.021)** | **(0.027)** | | | **(0.083)** | **(0.061)** | | |
| D.Shareh | | | **0.4833**\*\* | **0.5683**\* | | | **-0.6118**\*\* | **-0.6232**\*\* |
| | | | **(0.243)** | **(0.330)** | | | **(0.269)** | **(0.260)** |
| L. ROA | 0.6815\*\*\* | | | | 0.4336\*\*\* | | | |
| | (0.071) | | | | (0.120) | | | |
| L. ROE | | 0.2120\*\*\* | | | | | | |
| | | (0.051) | | | | | | |
| L2. ROA | | | -0.4575\*\*\* | | | | | |
| | | | (0.147) | | | | | |
| L2. ROE | | | | 0.6547\*\*\* | | | | -1.7999\*\*\* |
| | | | | (0.152) | | | | (0.156) |
| Grow | | | | | | 0.2096\*\*\* | | |
| | | | | | | (0.031) | | |
| AssLia | | | | | | | -1.6257\*\*\* | |
| | | | | | | | (0.462) | |
| LogAss | 0.0077\*\*\* | 0.0148\*\*\* | 0.0066\*\*\* | 0.0093\*\*\* | 0.0100\*\*\* | 0.0157\*\*\* | 0.0054\*\*\* | 0.0109\*\*\* |
| | (0.000) | (0.001) | (0.000) | (0.001) | (0.000) | (0.000) | (0.001) | (0.001) |
| Conc | -0.0316\*\* | -0.0100 | -0.0370 | 0.0790\*\*\* | -0.0607\*\*\* | -0.0206\*\*\* | -0.1103\*\*\* | -0.0760\*\*\* |
| | (0.015) | (0.007) | (0.023) | (0.014) | (0.015) | (0.005) | (0.031) | (0.013) |
| NonIns | -2.7673\*\* | 1.2689\*\*\* | -3.0164\* | -2.1187\*\*\* | -3.3992\*\*\* | 1.4382\*\*\* | 0.0770 | 4.2851\*\*\* |
| | (1.153) | (0.396) | (1.818) | (0.706) | (1.069) | (0.265) | (1.360) | (0.738) |
| Industry variables | Controlled | Controlled | Controlled | Controlled | Controlled | Controlled | Controlled | Controlled |
| Number of Samples N | 8715 | 8715 | 8715 | 8715 | 19500 | 19500 | 19500 | 19500 |

Note: The table employs the LIML regression approach. The symbols "\*," "\*\*," and "\*\*\*" in the table indicate that the statistic is significant at the 10%, 5%, and 1% levels, respectively. The alternative proxies of EPS, namely ROA and ROE, are the explanatory variables of the value creation model, while the lag terms of the shareholding ratio of insurance institutions (Shareh) are the corresponding explanatory variables, and the instrumental variable is the information degree of institutional investors' equity (Insfir3, Insfir5) and the lag terms of the corresponding explained variables; The difference terms D.ROA and D.ROE of ROA and ROE are the explained variables of the value creation model. The corresponding explanatory factors are the difference terms of the shareholding ratio of insurance institutions (D.Shareh). The instrumental variables are institutional investors' equity information degrees (Insfir3, Insfir5) and the explained variables' second-order lag terms.

of long-term holding type 2 IIIs is stronger in the value creation model, with the coefficient increasing from 0.0168 (column (1) of Table 5) to 0.0245 (column (3) of Table 8). Despite the decrease in significance level, it illustrates that the more IIIs whose purpose is to hold shares for a long time, the more motivated they are to participate in corporate supervision and corporate governance, resulting in a value-creating influence on the company they own. The FIFO type 2 still lacks a value creation function and has a detrimental impact. Although there are slight differences between the results in Table 8 and Tables 4 or 5, the direction and

**Table 8. Value selection and creation models with longer duration for long-term holding and shorter duration for FIFO.**

| | Value selection model (EPS) | | Value creation model (D.EPS) | |
|---|---|---|---|---|
| | Long-term holding 2 | FIFO 2 | Long-term holding 2 | FIFO 2 |
| | (1) | (2) | (3) | (4) |
| L.Shareh | **0.0048**** | **0.0170**** | | |
| | **(0.002)** | **(0.007)** | | |
| D.Shareh | | | 0.0245* | -0.0068** |
| | | | (0.013) | (0.003) |
| L.EPS | 0.5220*** | 0.5841*** | | |
| | (0.041) | (0.045) | | |
| L2.EPS | | | -0.2163*** | -0.2322*** |
| | | | (0.021) | (0.016) |
| Grow | 0.0003*** | 0.0002*** | 0.0002*** | 0.0002*** |
| | (0.000) | (0.000) | (0.000) | (0.000) |
| AssLia | 0.0003 | -0.0006 | 0.0006 | -0.0004 |
| | (0.000) | (0.000) | (0.000) | (0.000) |
| LogAss | -0.1070** | 0.0013 | -0.0936** | -0.0238 |
| | (0.047) | (0.025) | (0.046) | (0.017) |
| Conc | 0.0016* | 0.0001 | -0.0002 | -0.0002 |
| | (0.001) | (0.000) | (0.001) | (0.000) |
| NonIns | 0.0023*** | 0.0023*** | 0.0013*** | 0.0012*** |
| | (0.001) | (0.001) | (0.000) | (0.000) |
| Industry variables | Controlled | controlled | controlled | controlled |
| Joint significance | 33.42*** | 88.77*** | 15.03*** | 30.48*** |
| Under-identification | 170.042*** | 158.810*** | 51.411*** | 210.127*** |
| Weak instrumental variables (CD) | 388.447 | 406.527 | 30.646 | 1294.807 |
| Hensen J | 0.598 | 3.844 | 0.394 | 4.062 |
| Number of Samples N | 5754 | 14700 | 5754 | 14700 |

Note: In the table, the LIML regression approach is utilized. The joint significance of the time dummy variables is referred to as joint significance. The symbols "*," "**," and "***" in the table indicate that the statistic is significant at the 10%, 5%, and 1% levels, respectively. "In the first-stage regression, the instrumental variables utilized have no explanatory power for endogenous variables," is the null hypothesis of the under-identification test, and the value of the under-identification test in this work considerably rejects the null hypothesis. In the weak instrumental variable test table, only the CD statistic is presented. It is difficult to reject the null hypothesis—"the instrumental variable is significantly associated with the endogenous variable," that is, there is no weak instrumental variable problem, if the CD value is more than 10. The results of the over-identification test were reported by Hansen J. The number of instrumental variables in this study is more than the number of endogenous variables, meeting the over-identification test's prerequisite. The null hypothesis of "all instrumental factors are exogenous" is not rejected, and Hansen J's findings are not significant. It demonstrates the consistency with which model instrumental variables were chosen

significance of the coefficients are consistent with the assumptions and regression results of this research, which further verifies the robustness of the conclusions.

## Conclusion and policy recommendation

While there is a multitude of research on the influence of institutional investor holdings on business value, there is little research that focuses on IIIs exclusively. Studies on IIIs have become increasingly essential and meaningful with the growing importance of the insurance

industry in the financial sector, and the conflicts resulting from insurance funds owning shares in major Chinese real estate enterprises. This article examines the impact of IIIs on the companies they hold in a methodical and in-depth manner from the perspectives of value selection and creation. This study is separated into FIFO stock holdings and long-term stock holdings from the standpoint of value creation, and higher stock replacement frequency and longer stock holding periods. This article derives the following findings based on the quarterly data of listed businesses held by 1,654 insured institutions in China from 2012 to 2018: (1) Long-term equity-holding IIIs have both value selection and value creation functions; (2) the value creation function becomes more important among long-term stock-holding IIIs with an increase in the period during which they retain the company's shares; (3) FIFO IIIs have a value-inhibiting effect on the held company and serve a value selection role, rather than a value creation function.

The following aspects highlight the contributions of this paper: Theoretically, this paper examines the value creation of insurance companies from the perspectives of positive and negative motivation. It also examines the characteristics of insurance funds, insurance business, insurance industry chain extension, new accounting standards, and the impact of different stages of insurance company development on the value creation of insurance institutions. The extant literature on institutional investors' impact on company value is less focused on insurance companies. This work adds to the body of knowledge about institutional investors' heterogeneity. Additionally, theoretical analysis offers organizations with judgement criteria for determining the motives of insurance institution investment shareholders to some extent. The findings of this article give an empirical basis for regulators to encourage long-term holdings by insurance firms, differentiate the value creation function from the standpoint of investment horizon, and dispute the notion that IIIs are "barbarians". The study's findings give a reasonable foundation for businesses to deal with IIIs' holdings correctly. The empirical findings suggest that a held corporation can be exposed to IIIs, particularly long-term IIIs. However, for FIFO IIIs, it is vital to protect the company's value from unfavorable effects.

This study makes the following practical recommendations based on a theoretical and empirical analysis of the entire material. First, incentive mechanisms and policy measures should be implemented by regulatory authorities to encourage insurance companies to hold long-term shares. Long-term insurance company ownership of high-quality enterprises is a win-win situation with favorable externalities. Our findings show that long-term shareholding can meet the long-term liabilities characteristics of insurance funds, meet the extension of the insurance industry chain, adapt to new accounting standards to make investment performance more stable, avoid frequent selection of corporate shareholding, and reduce labor costs and risks for insurance companies. Additionally, it can increase the value of a company, according to the empirical analysis in this paper. Furthermore, encouraging insurance companies to hold long-term shares will result in beneficial capital market externalities. The lack of long-term consistent capital flows in the capital market contributes to the degree of stock market volatility. Long-term shareholding by insurance companies will help to smooth out the stock market's severe volatility. However, appropriate regulatory policies are required to promote insurance's role as a societal "stabilizer." Equity participation requirement for IIIs being lowered even more may lead to the encouragement of value investing, and long-term investing and participation in corporate governance. Furthermore, laws and regulations governing equity investment, as well as the information disclosure system, should be improved to reduce speculation among small shareholders and other investors, prevent shareholders and managers from colluding to artificially raise stock prices, and limit potential speculative behavior at the legal system level. It is more beneficial to increasing the company's worth and allow it to develop in a sustainable manner.

Second, privately held companies must avoid perceiving insurance institutions as "savages". The company's shareholding structure should be given top priority by businesses, and appropriate steps must be taken to ensure that the ownership structure of the company is not overly dispersed. The most fundamental and effective strategy to control the corporation is to maintain a relatively favorable shareholding. The company's articles of association and system, board of directors and independent director system, and decision-making process must all be upgraded to avoid hostile takeovers by institutional investors. Finally, corporate management should work together with insurance institutional investors, approach different insurance institutional investors differently, assess whether they have positive value creation goals, and make decisions that benefit the company's value. They should pay close attention to changes in shareholdings in the capital market and avoid excessive group speculation driven by short-term insurance institutional investors to protect the company's long-term strategic worth. This article's focus on the impact of Chinese insurance institutions' shareholdings on firm value can be useful as a benchmark for other nations, particularly emerging economies.

The study has the following shortcomings, flaws, and future research directions. First, this paper examines the IIIs on their own, while considering the diversity of institutional investors. However, the impact of shareholding by different types of insurance companies on business value is not examined further. How does shareholding by life and property insurers effect the company's value differently? What is the difference between small and medium insurers, and large insurance groups when it comes to the influence of equity holdings on company value? Do insurance companies at different stages of development have different effects on the value of the held company? The classification research of insurance businesses may provide regulatory agencies and companies with more information and assistance. Second, while this article indicates that long-term shareholding by insurance institutions can increase a company's value, it does not go into detail on how this can be done. Does shareholding by IIIs increase the company's value via strengthening internal governance, lowering agency costs, enhancing business synergy, or in other ways? These kind of path analyses can help insurers and stakeholders collaborate more effectively. Third, the types of companies in which insurance companies own shares are not distinguished in this article. Which sort of company's value grows faster and which grows slower for insurance institutions' long-term holdings? Should IIIs concentrate their efforts on holding companies that collaborate with the insurance industry? Is there a difference in the impact of an insurance company's shareholding on a profitable or loss-making business? Such corporate heterogeneity research could serve as a guide and aid to insurance organizations to utilize their money better and select appropriate shareholding objectives. Fourth, it is also interesting and meaningful to observe the performance changes of value selection and value creation under different insurance institutions' shareholding ratios by using quantile grouping. Our results hold up for only observations with positive EPS, ROA or ROE. Sometimes when companies are going through serious difficulties their normal "roles or priorities" change.

Furthermore, since our data comes from China, what are the variances in the impact of insurance institutional investors on corporate value in different countries? What are the causes of these discrepancies? The questions above are additional in-depth inquiries that we did not address in this article. These and other related concerns are research topics that need to be addressed in the future.

## Supporting information

**S1 Data.**
(ZIP)

## Acknowledgments

We would like to acknowledge the valuable suggestions of the editor and anonymous reviewers. We also give sincere gratitude to Dr. Weiwei Cui (Southwestern University of Finance and Economics) and Shuangrui Han (Southwestern University of Finance and Economics) for their devoted work and data processing for this manuscript.

## Author Contributions

**Conceptualization:** Xing Rong, Tingting Zhang, Kai Liu.

**Formal analysis:** Xing Rong.

**Methodology:** Tingting Zhang.

**Writing – original draft:** Kai Liu.

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
