## [Decision Letter · Decision Letter 0]

10 Jan 2022

PONE-D-21-31055The impact of insurance institutional investors on corporation value from selection and creation perspectivePLOS ONE

Dear Dr. Liu,

Thank you for submitting your manuscript to PLOS ONE. After careful consideration, I feel that it has merit but does not fully meet PLOS ONE’s publication criteria as it currently stands. Therefore, I invite you to submit a revised version of the manuscript that addresses the points raised during the review process. I agree with the reviewers that there are many major concerns that need to be addressed before considering the paper for publication. The introduction section does not provide enough information to justify the research of this paper. The contribution and the findings must be clarified.

On the other hand, the importance of this topic as well as the limitation to China must be also clarified. Probably it should be interesting to attend the reviewers one suggestion about this point.

I find also that the literature review section is not properly organized, so it is difficult to follow and to understand.

Other major concerns are referred to the methodological section of the manuscript. There is not a clear justification for why this methodology is used or the variables were selected.

Finally, I agree with the reviewers that the conclusion section needs to be improved. The results must be compared with other previous findings of the literature on this topic.

We look forward to receiving your revised manuscript.

Kind regards,

J E. Trinidad Segovia

Academic Editor

PLOS ONE

Journal Requirements:

Reviewers' comments:

Reviewer's Responses to Questions

**Comments to the Author**

1. Is the manuscript technically sound, and do the data support the conclusions?

Reviewer #1: Partly

Reviewer #2: Partly

2. Has the statistical analysis been performed appropriately and rigorously? 

Reviewer #1: Yes

Reviewer #2: No

3. Have the authors made all data underlying the findings in their manuscript fully available?

Reviewer #1: Yes

Reviewer #2: Yes

4. Is the manuscript presented in an intelligible fashion and written in standard English?

Reviewer #1: Yes

Reviewer #2: No

5. Review Comments to the Author

Reviewer #1: Thank you very much for giving me the opportunity to review the article ‘The impact of insurance institutional investors on corporation value from selection and creation perspective’. This study seeks to examine the influence of insurance institutional investors on company value selection and creation. While interesting, I have several important concerns, which I explain following the structure of the manuscript:

1. Abstract

I suggest author(s) make an extra effort to write a more attractive abstract. It would be interesting to at least state the main objective of the study and highlight the most important findings. For example, you state that ‘The impact that insurance institutional investors have on corporations is an urgent problem addressed in academic and practical research’, but why it is an urgent problem? In my opinion, this is not clear in the summary, nor in the full article.

2. Introduction

The introduction is extremely poor. It is not clearly explained why this study is necessary. The goals pursued are also unclear. Until the ‘Research Hypothesis’ section, it did not seem very clear to me what specific goals the author(s) are addressing in this article. I encourage author(s) to explicitly state the research goals/questions and develop the motivation for why it is necessary to carry them out. To this end, I suggest the following:

1. Rewrite the introduction following the traditional structure of 2 pages and 4 paragraphs: (1) review of what we know about the research topic; (2) identification of research gaps that your research fills; (3) research objectives and brief description of how author intends to achieve those objectives; and (4) contributions to the literature.

2. Include calls from scholars for research on this topic and explain why these scholars have called for research on this topic.

The paper focuses on China, but why China? Justifying the context in which your study takes place, as well as explaining why this context is appropriate for analysing your research question, can improve the quality of the manuscript and help readers better understand the results.

Why is it important to analyse the impact of institutional insurance investors on value selection and creation and no other types of investors? What differentiates them from other investors with respect to these issues? Are there any particularities that make their study interesting in terms of value selection and creation?

As the introduction should be reviewed to strengthen motivation, similar changes should be considered in the conclusion to provide a more accurate picture of how your study makes a value-added contribution to the literature and how your study offers some practical implications.

3. Literature review, Theoretical Analysis and Hypothesis

This section is to some extent confusing. I see a lack of common thread, as each subsection does not link with the rest, and the hypotheses are very poorly justified. I suggest the following structure:

1. A first subsection dedicated to insurance institutional investors, their conceptualisation, their main and most important differences with respect to other investors, their singular impact on company value selection and creation, etc. In my opinion, the inclusion of this subsection would be extremely important, as insurance institutional investors, as a concept, is not properly defined throughout the paper.

2. A second subsection which reviews prior literature directly related to the topic, and explains how such studies give rise to the present work.

3. A third subsection focusing on the effect of insurance institutional investors on company value selection and creation. This section should be devoted to providing more convincing and substantial arguments to justify how insurance institutional investors can affect company value selection and creation.

Finally, the research goals should be placed in the introduction section rather than in the Theoretical Analysis and Hypothesis section. As mentioned, this part should focus on providing stronger theoretical arguments to support the hypotheses under study.

4. Methodology

I encourage the author(s) to explain better why they have chosen the regression techniques used and not others. For example, why not SEM?

5. Empirical results

The authors should make a greater effort to find previous papers that confirms their results.

6. Conclusions

In this section, I really missed a deep reflection on the part of the author(s). As researchers, our work should reflect our ability to think and not just describe results. This paper reveals interesting results, and the importance of these results is not adequately reflected in this section. It would be interesting, for example, to relate the obtained results more closely to previous literature and to point out more explicitly the contributions of the study in comparison to the previous literature. In addition, it is very important to elaborate further on the contributions and practical implications, as well as to indicate some limitations and future avenues of research may help to guide future researchers in this field and improve the quality of the manuscript. For example, this section could also be strengthened by further detailing the practical implications of the study, e.g. for Chinese companies.

Other minor issues

-There are no references to PLOS ONE. Thus, I suggest authors to include some article from PLOS ONE to underline the importance of the topic for the journal.

In summary, this paper has some potential, but requires considerable improvement before it can be published. I hope that these comments will be helpful in improving the manuscript. Best wishes for the future!

Reviewer #2: I appreciate the opportunity to renew your manuscript entitled “The impact of insurance institutional investors on corporation value from selection and creation perspective”. The purpose of the paper is the impact that insurance institutional investors may have on corporations. I believe that you are exploring a interesting topic. From my point of view, your paper is promising although needs mayor improvements I hope you find them helpful.

Main Comments

The whole paper needs a English grammar review.

1. INTRODUCTION

I miss in this section the following aspects: gap in the literature that your paper aims to fill in, your contribution and findings. In this way, I think it is necessary to distinguish between this topic worldwide and specifically in China.

I suggest restructuring Sections “Introduction”, “Literature Review” and “Theoretical Analysis and Hypothesis”.

From lines 50 to 54 I think you need to add citations to support those statements.

The justification of you paper, lines 60-61 is quite weak.

2. LITERATURE REVIEW / THEORETICAL ANALYSIS / HYPOTHESES

As stated before, I believe you should restructure this section, linking the review of the literature to the hypothesis. Indeed, the hypotheses should be right after the literature review of each of them, as usual. There are a lot of information, but it is disconnected and its reading is quite messy, and sometimes, apparently, contradictory.

You show the arguments in the literature in favour and against the impact of insurance institutional investors on companies’ return. However, you do not take a position in your paper. Moreover, your review looks a kind of disconnected from the hypotheses. That is why I suggest restructuring the three first sections.

In line 73 you mention “information disclosure”, however I cannot see the link of this concept with your paper.

In line 131 is the first time you use the acronym CSR, and it is necessary to explain what these words stand for.

Citation in line 143 is wrong.

In lines 193-194 you state that you contribute to the literature analysing positive and negative motives, but I miss them in your paper.

In lines 216-217 you need to use citations to support such a strong assumption.

Lines 222-223 and 262 seem to be contradictory, as line 254 and 257-258.

In line 276, in your H1, suddenly, you use the term FIFO. You need to explain this concept and how it is connected to the literature and the aim of you paper before stating the hypothesis.

3. MODEL DESIGN AND VARIABLES

I strongly encourage you to re-write “Main Model Design” in a clearer way. It is difficult to understand your models.

In line 294, what is “issue t“? It means time, but you use different words.

In line 296, “control variables” instead of “control variable”.

Why are you using “L”, “L2” and “D”? Explain the economic reasoning and connect it to the hypotheses.

It is necessary to support with citations the different dependent variables shown in this section.

Regarding the endogeneity problem, in my case, it is the first time that I meet models like yours to deal with this potential problem. I believe either you strongly support your models (from A to D) with the literature or use the well accepted (system) GMM approach, which uses most of all the available moments (lags) for the independent and control variables. In your case, you only use the second lag (L2).

I believe that it could be useful to add in Table 2 and 3, variables used in your paper such as: InsFir3, InsFir5, InsDura.

4. DATA PROCESSING AND DESCRIPTIVE STATISTICS

A better explanation of the three requirements you mentioned in line 388.

Your sample started from 2012 to 2018 (25 quarters) but in line 411 you use data from 2010 to 2018 (33, segments). A better explanation is needed.

It is important to specify whether you are using a balanced or unbalanced panel. I think, reading your data in the regression tables that it is balanced, but you need you said clearer. But, if so, if it is balanced, you need to take care of the so-called “survival bias” effect in panel data set.

In line 407, you use the terms “ST” and “*ST”, again two new concepts suddenly shown in the paper. What do they stand for?

From line 418 to 416, what is it the “conversion” between x-times (e.g. 1.4 times) and quarters? Besides, why you select the criterium of “more than 11 quarters and less than 4”? Please, explain and support with the literature.

You must improve your description of the control variables. You does not describe all of them. What is “agency variable”? (again, suddenly, you use a new concept).

In Table 2, have you checked whether companies with many negative values in the dependent variables behave differently than companies with positive? Could it be possible to show a different behaviour? How have you treated outliers?

5. MODEL REGRESSION ANALYSIS AND CONTERFACTUAL TEST

Table 4 and 5 do not show the same format.

At the bottom of the regression tables, I suggest using the name of the tests applied (Hansen, Sargan), it could help.

In Table 6, are you using LIML or 2SLS?

Is there a connection between Table 6 and the hypotheses?

6. FURTHER STUDY

Why is the support (citations) used in the new “classification criteria” shown in this section?

In line 549, I might be wrong, but I think that the significance level does not improve as you stated. In line 557 I believe that “0.00463” is wrong, and the right number is “0.017”. In any case, explain better which tables you are comparing to avoid any misunderstanding.

In lines 563-564, what does it add this new “perspective”? Why?

Again, in Table 7, (Why 7.1?), are you using LIML or 2SLS?

7. CONCLUSION AND POLICY RECOMMENDATION

I miss in this section “limitations” and “implications” of your findings

6. PLOS authors have the option to publish the peer review history of their article (what does this mean?). If published, this will include your full peer review and any attached files.

Reviewer #1: No

Reviewer #2: No

---

## [Author Response · Author response to Decision Letter 0]

25 Mar 2022

Reviewer 1: We have incorporated all of your suggestions into our revision. Your constructive comments on the paper were extremely valuable in improving the quality of the paper. Thank you very much.

Reviewer 2: We have incorporated all of your suggestions into our revision. Your constructive comments on the paper were extremely valuable in improving the quality of the paper. Thank you very much.

---

## [Decision Letter · Decision Letter 1]

22 Apr 2022

PONE-D-21-31055R1The Impact of Insurance Institutional Investors on Corporate Value from Selection and Creation PerspectivePLOS ONE

Dear Dr. Liu,

Thank you for submitting your manuscript to PLOS ONE. After careful consideration, I feel that it has merit but does not fully meet PLOS ONE’s publication criteria as it currently stands. Therefore, I invite you to submit a revised version of the manuscript that addresses the points raised during the review process.

You will see from the referees’ comments that additional information needs to be provided, in regard to methodology and model design and variables, and I ask that this be provided before we consider your manuscript further.

We look forward to receiving your revised manuscript.

Kind regards,

J E. Trinidad Segovia

Section Editor

PLOS ONE

Journal Requirements:

Reviewers' comments:

Reviewer's Responses to Questions

**Comments to the Author**

1. If the authors have adequately addressed your comments raised in a previous round of review and you feel that this manuscript is now acceptable for publication, you may indicate that here to bypass the “Comments to the Author” section, enter your conflict of interest statement in the “Confidential to Editor” section, and submit your "Accept" recommendation.

Reviewer #1: All comments have been addressed

Reviewer #2: All comments have been addressed

2. Is the manuscript technically sound, and do the data support the conclusions?

Reviewer #1: Yes

Reviewer #2: Yes

3. Has the statistical analysis been performed appropriately and rigorously? 

Reviewer #1: Yes

Reviewer #2: Yes

4. Have the authors made all data underlying the findings in their manuscript fully available?

Reviewer #1: No

Reviewer #2: Yes

5. Is the manuscript presented in an intelligible fashion and written in standard English?

Reviewer #1: Yes

Reviewer #2: Yes

6. Review Comments to the Author

Reviewer #1: Dear Author(s),

Thank you very much for the opportunity to review the R1 version of your manuscript. I would like to acknowledge your efforts to address my concerns with the previous version, as well as the other reviewer's comments and suggestions. I appreciate many positive aspects in the development of the manuscript. Below, I propose some suggestions that should be considered before publication:

- The paper would benefit (in my view) if you put the Sample Description section first, and then all the content related to Model Design and Variables.

- References should be ordered throughout the article according to PLOS ONE guidelines. That is, the first reference in the paper should be number 1, the next number 2, and so on.

Good luck with your research!

Reviewer #2: I really appreciate that my comments have helped to improve your paper, and I am glad that you have taken them into consideration. And I appreciate all the effort you have done.

Again, just to keep improving a little bit more the paper, new comments can be found below.

1. ABSTRACT

I recommend not to be so categorical when you state that “this study rectifies”. I could be much better to use a softer or smoother expression, such as “shed light”, “provides more insight…”.

2. INTRODUCTION

Although in the Abstract you explain what a IIIs is, I think that it is necessary to explain it again in this section.

When you start describing the three roles (line 53) of institutional investors, I think that you need to help readers to distinguish these three roles in the following paragraphs. You me it took time to distinguish them.

In lines 136-138 you repeat the “Vanke-Baoneng Battle” issue as you did in lines 45-47 using almost the same expression.

3. LITERATURE REVIEW / THEORETICAL ANALYSIS / HYPOTHESES

From my point of view a brief definition of the two types of investment roles would be welcome: value selection and value creation.

Regarding the three hypotheses, and to make you think for future papers, it would be interesting to analyse whether these hypotheses varies as the percentage of shares (Shareh variable) also varies, using for example, quartiles.

What is the added value of H2 compared to the empirical evidence of references 10, 44, 72 and 73? If this papers confirm that short-term IIIs’ stock investments cannon improve company performance, what are you added to the literature? Explain a little bit.

4. MODEL DESIGN AND VARIABLES

Citations from the literature justifying or supporting the “Model design for counterfactual testing” would be also welcome.

In your answers you mention that your paper follow Arellano and Bond (1991) but I cannot see the citation in the paper.

Regarding GMM, I really appreciate the effort you have done calculating and showing the outcome from GMM. In my opinion, GMM can be used unless there is a justification that the control variable cannot have potential problems of endogeneity. But you can keep the GMM outcome out of the paper, and available under request since the results almost are the same.

5. DATA PROCESSING AND DESCRIPTIVE STATISTICS

Again, to improve the paper or for a new one, I think that you could test whether your results hold up or change for only observations with positive EPS, ROA or ROE. Sometimes when companies are going through serious difficulties their normal “roles or priorities” change.

In Table 3, coefficient between Grown and Conc, there are five digits after dot (0.00300).

6. MODEL REGRESSION ANALYSIS AND CONTERFACTUAL TEST

In Table 5 you forgot to add the “Note” statements.

7. PLOS authors have the option to publish the peer review history of their article (what does this mean?). If published, this will include your full peer review and any attached files.

Reviewer #1: No

Reviewer #2: No

---

## [Author Response · Author response to Decision Letter 1]

17 May 2022

Reviewer 1: We have incorporated all of your suggestions into our revision. They were very helpful. Thank you very much for your time.

Reviewer 2: We have incorporated all of your suggestions into our revision. They were very helpful. Thank you very much for your time.

---

## [Decision Letter · Decision Letter 2]

24 May 2022

The Impact of Insurance Institutional Investors on Corporate Value from Selection and Creation Perspective

PONE-D-21-31055R2

Dear Dr. Liu,

We’re pleased to inform you that your manuscript has been judged scientifically suitable for publication and will be formally accepted for publication once it meets all outstanding technical requirements.

Kind regards,

J E. Trinidad Segovia

Section Editor

PLOS ONE

Additional Editor Comments (optional):

Reviewers' comments:

Reviewer's Responses to Questions

**Comments to the Author**

1. If the authors have adequately addressed your comments raised in a previous round of review and you feel that this manuscript is now acceptable for publication, you may indicate that here to bypass the “Comments to the Author” section, enter your conflict of interest statement in the “Confidential to Editor” section, and submit your "Accept" recommendation.

Reviewer #1: All comments have been addressed

2. Is the manuscript technically sound, and do the data support the conclusions?

Reviewer #1: Yes

3. Has the statistical analysis been performed appropriately and rigorously? 

Reviewer #1: Yes

4. Have the authors made all data underlying the findings in their manuscript fully available?

Reviewer #1: No

5. Is the manuscript presented in an intelligible fashion and written in standard English?

Reviewer #1: Yes

6. Review Comments to the Author

Reviewer #1: (No Response)

7. PLOS authors have the option to publish the peer review history of their article (what does this mean?). If published, this will include your full peer review and any attached files.

Reviewer #1: No

---

## [Editor Report · Acceptance letter]

14 Jun 2022

PONE-D-21-31055R2 

The Impact of Insurance Institutional Investors on Corporate Value from Selection and Creation Perspective 

Dear Dr. Liu:

I'm pleased to inform you that your manuscript has been deemed suitable for publication in PLOS ONE. Congratulations! Your manuscript is now with our production department. 

Kind regards, 

on behalf of

Dr. J E. Trinidad Segovia 

Section Editor

PLOS ONE